# Relative gradient optimization of the Jacobian term in unsupervised deep learning

**Luigi Gresele**[*1,2]  **Giancarlo Fissore**[*3,4]  **Adrián Javaloy** [1]

**Bernhard Schölkopf** [1]  **Aapo Hyvärinen** [3,5]

[1]Max Planck Institute for Intelligent Systems, Tübingen, Germany
[2]Max Planck Institute for Biological Cybernetics, Tübingen, Germany
[3] Université Paris-Saclay, Inria, Inria Saclay-Île-de-France, 91120, Palaiseau, France
[4] Université Paris-Saclay, CNRS, Laboratoire de recherche en informatique, 91405, Orsay, France
[5] Dept of Computer Science, University of Helsinki, Finland
`luigi.gresele@tuebingen.mpg.de; giancarlo.fissore@inria.fr`

## Abstract

Learning expressive probabilistic models correctly describing the data is a ubiquitous problem in machine learning. A popular approach for solving it is mapping the observations into a representation space with a simple joint distribution, which can typically be written as a product of its marginals — thus drawing a connection with the field of nonlinear independent component analysis. Deep density models have been widely used for this task, but their maximum likelihood based training requires estimating the log-determinant of the Jacobian and is computationally expensive, thus imposing a trade-off between computation and expressive power. In this work, we propose a new approach for exact training of such neural networks. Based on relative gradients, we exploit the matrix structure of neural network parameters to compute updates efficiently even in high-dimensional spaces; the computational cost of the training is quadratic in the input size, in contrast with the cubic scaling of naive approaches. This allows fast training with objective functions involving the log-determinant of the Jacobian, without imposing constraints on its structure, in stark contrast to autoregressive normalizing flows.

## 1   Introduction

Many problems of machine learning and statistics involve learning invertible transformations of complex, multimodal probability distributions into simple ones. One example is density estimation through latent variable models under a specified base distribution [51], which can also have applications in data generation [14, 33, 19] and variational inference [44]. Another example is nonlinear independent component analysis (nonlinear ICA), where we want to extract simple, disentangled features out of the observed data [27, 30].

One approach to learn such transformations, introduced in [50] in the context of density estimation, is to represent them as a composition of simple maps, the sequential application of which enables high expressivity and a large class of representable transformations. Deep neural networks parameterize functions of multivariate variables as modular sequences of linear transformations and component-wise activation functions, thus providing a natural framework for implementing that idea, as already proposed in [45].

---

[*]Equal contribution

Unfortunately, however, typical strategies employed in neural networks training do not scale well for objective functions like the aforementioned ones; in fact, through the change of variable formula, the logarithm of the absolute value of the determinant of the Jacobian appears in the objective. Its exact computation, let alone its optimization, quickly gets prohibitively computationally demanding as the data dimensionality grows.

A large part of the research on deep density estimation, generally referred to under the term *autoregressive normalizing flows*, has therefore been dedicated to considering a restricted class of transformations such that the computation of the Jacobian term is trivial [14, 44, 15, 34, 25, 12], thus imposing a tradeoff between computation and expressive power. While such models can approximate arbitrary probability distributions, the extracted features are strongly restricted based on the imposed triangular structure, which prevents the system from learning a properly disentangled representation. Other strategies involve the optimization of an approximation of the exact objective [5], and continuous-time analogs of normalizing flows for which the likelihood (or some approximation thereof) can be computed using relatively cheap operations [13, 19].

In this work, we provide an efficient way to optimize the exact maximum likelihood objective for deep density estimation as well as for learning disentangled representations by latent variable models. We consider a nonlinear, invertible transformation from the observed to the latent space which is parameterized through fully connected neural networks. The weight matrices are merely constrained to be invertible. The starting point is that the parameters of the linear transformations are matrices; this allows us to exploit properties of the Riemannian geometry of matrix spaces to derive parameter updates in terms of the relative gradient, which was originally introduced as the natural gradient in the context of linear ICA [11, 2], and which can be feasibly computed. We show how this can be integrated with the usual backpropagation employed to compute gradients in neural network training, yielding an overall efficient way to optimize the Jacobian term in neural networks. This is a general optimization approach which is potentially useful for any objective involving such a Jacobian term, and is likely to find many applications in diverse areas of probabilistic modelling, for example in the context of Bayesian active learning for the computation of the information gain score [48], or for fitting the reverse Kullback-Leibler divergence in variational inference [54, 7].

The computational cost of our proposed optimization procedure is quadratic in the input size— essentially the same as ordinary backpropagation— which is in stark contrast with the cubic scaling of the naive way of optimizing via automatic differentiation. The joint asymptotic scaling of forward and backward pass as a function of the input size is therefore the same that aforementioned alternative methods achieve by imposing strong restrictions on the neural network structure [44] and thus on the class of functions they can represent. In contrast, our approach allows to efficiently optimize the exact objective for neural networks with arbitrary Jacobians.

In sections 2 and 3 we review maximum likelihood estimation for latent variable models, backpropagation and the Jacobian term for neural networks, and discuss the complexity of the naive approaches for optimizing the Jacobian term. Then in section 4 we discuss the relative gradient, and show how it can be integrated with backpropagation resulting in an efficient procedure. We verify empirically the computational speedup our method provides in section 5.

## 2 Background

### 2.1 Maximum likelihood for latent variable models

Consider a generative model of the form

$$\mathbf{x} = \mathbf{f}(\mathbf{s}) \tag{1}$$

where $\mathbf{s} \in \mathbb{R}^D$ is the latent variable, $\mathbf{x} \in \mathbb{R}^D$ represents the observed variable and $\mathbf{f} : \mathbb{R}^D \to \mathbb{R}^D$ is a deterministic and invertible function, which we refer to as *forward* transformation. Under the model specified above, the log-likelihood of a single datapoint $\mathbf{x}$ can be written as

$$\log p_{\boldsymbol{\theta}}(\mathbf{x}) = \log p_s(\mathbf{g}_{\boldsymbol{\theta}}(\mathbf{x})) + \log |\det \mathbf{J}\mathbf{g}_{\boldsymbol{\theta}}(\mathbf{x})|, \tag{2}$$

where $\mathbf{g}_{\boldsymbol{\theta}}$ is some representation with parameters $\boldsymbol{\theta}$ of the *inverse* transformation[2] of $\mathbf{f}$; $\mathbf{J}\mathbf{g}_{\boldsymbol{\theta}}(\mathbf{x}) \in \mathbb{R}^{D \times D}$ its Jacobian computed at the point $\mathbf{x}$, whose elements are the partial derivatives

$[\mathbf{Jg_\theta}(\mathbf{x})]_{ij} = \partial g_\theta^i(\mathbf{x})/\partial x^j$; and $p_\theta$ and $p_s$ denote, respectively, the probability density functions of $\mathbf{x}$ and of the latent variable $\mathbf{s}$ under the specified model. In many cases, it is additionally assumed that the distribution of the latent variable is sufficiently simple; for example, that it factorizes in its components,

$$\log p_\theta(\mathbf{x}) = \sum_i \log p_i(\mathbf{g}_\theta^i(\mathbf{x})) + \log |\det \mathbf{Jg_\theta}(\mathbf{x})| \,. \tag{3}$$

In this case, the problem can be interpreted as nonlinear independent component analysis (nonlinear ICA), and the components of $\mathbf{g}_\theta(\mathbf{x})$ are estimates of the original sources $\mathbf{s}$.

Another variant of this framework can be developed to solve the problem that nonlinear ICA is, in general, not identifiable without additional assumptions [29]; that means, even if the data is generated according to the assumed model, there is no guarantee that the recovered sources bear any simple relationship to the true ones. In order to obtain identifiability, it is possible to consider models [27, 28, 30, 20] in which the latent variables are not *unconditionally* independent, but rather *conditionally* independent given an additional, observed variable $\mathbf{u} \in \mathbb{R}^d$,

$$\log p_\theta(\mathbf{x}|\mathbf{u}) = \sum_i \log p_i(\mathbf{g}_\theta^i(\mathbf{x})|\mathbf{u}) + \log |\det \mathbf{Jg_\theta}(\mathbf{x})| \,, \tag{4}$$

where $d$ can be equal to or different from $D$ depending on the model.

Maximum likelihood estimation for the model parameters amounts to finding, through optimization, the parameters $\theta^*$ such that the expectation of the likelihood given by the expression in equation (3) is maximized. For all practical purposes, the expectation will be substituted with the sample average. Specifically, for optimization purposes, we will be interested in the computation of a gradient of such term on mini-batches of one or few datapoints, such that stochastic gradient descent can be employed.

## 2.2 Neural networks and backpropagation

Neural networks provide a flexible parametric function class for representing $\mathbf{g}_\theta$ through a sequential composition of transformations, $\mathbf{g}_\theta = \mathbf{g}_L \circ \ldots \circ \mathbf{g}_2 \circ \mathbf{g}_1$, where $L$ defines the number of layers of the network. When an input pattern $\mathbf{x}$ is presented to the network, it produces a final output $\mathbf{z}_L$ and a series of intermediate outputs. By defining $\mathbf{z}_0 = \mathbf{x}$ and $\mathbf{z}_L = \mathbf{g}_\theta(\mathbf{x})$, we can write the forward evaluation as

$$\mathbf{z}_k = \mathbf{g}_k(\mathbf{z}_{k-1}) \ \text{ for } k = 1, \ldots, L \,. \tag{5}$$

Each module $\mathbf{g}_k$ of the network involves two transformations,

(a) a coupling layer $C_{\mathbf{W}_k}$, that couples the inputs to the layer with the parameters $\mathbf{W}_k$ to optimize;

(b) other arbitrary manipulations $\boldsymbol{\sigma}$ of inputs/outputs. Typically, these are element-wise non-linear activation functions with fixed parameters; we can for simplicity think of them as operations of the form $\boldsymbol{\sigma}(\mathbf{x}) = (\sigma(x_1), \ldots, \sigma(x_n))$ applied to vector variables.

The resulting transformation can thus be written as $\mathbf{g}_k(\mathbf{z}_{k-1}) = \boldsymbol{\sigma}(C_{\mathbf{W}_k}(\mathbf{z}_{k-1}))$.

We will focus on fully connected modules, where the coupling $C_{\mathbf{W}}$ is simply a matrix-vector multiplication between the weights $\mathbf{W}_k$ and the input to the $k$-th layer; overall, the transformation operated by such a module can be expressed as $\boldsymbol{\sigma}(\mathbf{W}_k \mathbf{z}_{k-1})$. Another kind of coupling layer is given by convolutional layers, typically used in convolutional neural networks [36].

The parameters of the network are randomly initialized and then learned by gradient based optimization with an objective function $\mathcal{L}$, which is a scalar function of the final output of the network. At each learning step, updates for the weights are proportional to the partial derivative of the loss with respect to each weight.

The computation of these derivatives is typically performed by backpropagation [47], a specialized instance of automatic differentiation. Backpropagation involves a two-phase process. Firstly, during a *forward pass*, the intermediate and final outputs of the network $\mathbf{z}_1, \ldots, \mathbf{z}_L$ are evaluated and a value for the loss is returned. Then, in a second phase termed *backward pass*, derivatives of the loss with respect to each individual parameter of the network are computed by application of the chain rule. The gradients are computed one layer at a time, from the last layer to the first one; in the process,

the intermediate outputs of the forward pass are reused, employing dynamic programming to avoid redundant calculations of intermediate, repeated terms.[3]

In matrix notation, the updates for the weights of the $k$-th fully connected layer $\mathbf{W}_k$ can then be written as

$$\Delta \mathbf{W}_k \propto \mathbf{z}_{k-1} \boldsymbol{\delta}_k^\top \ , \tag{6}$$

where $\boldsymbol{\delta}_k$ is the cumulative result of the backward computation in the backpropagation step up to the $k$-th layer, also called backpropagated error. We report the full derivation in appendix A. We adopt the convention of defining $\mathbf{x}$, $\mathbf{z}_k$ and $\boldsymbol{\delta}_k$ as column vectors.

### 2.3 Difficulty of optimizing the Jacobian term of neural networks

In the case of the objective function specified in Eq. (3), we have $\mathcal{L}(\mathbf{x}) = \log p_{\boldsymbol{\theta}}(\mathbf{x})$. By defining

$$\mathcal{L}_p(\mathbf{x}) = \sum_i \log p_i(\mathbf{g}_{\boldsymbol{\theta}}^i(\mathbf{x})); \quad \mathcal{L}_J(\mathbf{x}) = \log |\det \mathbf{J} \mathbf{g}_{\boldsymbol{\theta}}(\mathbf{x})| \ , \tag{7}$$

the objective can be rewritten as $\mathcal{L}(\mathbf{x}) = \mathcal{L}_p(\mathbf{x}) + \mathcal{L}_J(\mathbf{x})$. The evaluation of the gradient of the first term $\mathcal{L}_p$ can be performed easily if a simple form for the latent density is chosen, as it only requires simple operations on top of a single forward pass of the neural network. Given that the loss is a scalar, as backpropagation is an instance of reverse mode differentiation [4], backpropagating the error relative to it in order to evaluate the gradients does not increase the overall complexity with respect to the forward pass alone.

In contrast, the evaluation of the gradient of the second term, $\mathcal{L}_J$, is very problematic, and our main concern in this paper. The key computational bottleneck is in fact given by the evaluation of the Jacobian during the forward pass. Since the Jacobian involves derivatives of the function $\mathbf{g}_{\boldsymbol{\theta}}$ with respect to its inputs $\mathbf{x}$, this evaluation can again be performed through automatic differentiation. Overall, it can be shown [4] that both forward and backward mode automatic differentiation for a $L$-layer, fully connected neural network scale as $\mathcal{O}(LD^3)$, with $L$ the number of layers. This is prohibitive in many practical applications with a large data dimension $D$.

**Normalizing flows with simple Jacobians** An approach to alleviate the computational cost of this operation is to deploy special neural network architectures for which the evaluation of $\mathcal{L}_J$ is trivial. For example, in autoregressive normalizing flows [14, 15, 34, 25] the Jacobian of the transformation is constrained to be lower triangular. In this case, its determinant can be trivially computed with a linear cost in $D$. Notice however that the computational cost of the forward pass still scales quadratically in $D$; the overall complexity of forward plus backward pass is therefore still quadratic in the input size [44].

Most critically, such architectures imply a strong restriction on the class of transformations that can be learned. While it can be shown, based on [29], that under certain conditions this class of functions has universal approximation capacity for *densities* [25], that is less general than other notions of universal approximation [23, 24]. In fact it is obvious that functions with such triangular Jacobians cannot be universal approximators of *functions*, since, for example, the first variable can only depend on the first variable. This is a severe problem in learning features for disentanglement, for example by nonlinear ICA [27, 30], which would usually require unconstrained Jacobians. In other words, such restrictions might imply that the deployed networks are not general purpose: [5] showed that constrained designs typically used for density estimation can severely hurt discriminative performance. We further elaborate on this point in appendix E. Note that fully connected modules have elsewhere been termed *linear* flows [42], and are a strict generalization of autoregressive flows.[4]

## 3 Log-determinant of the Jacobian for fully connected neural networks

As a first step toward efficient optimization of the $\mathcal{L}_J$ term, we next provide the explicit form of the Jacobian for fully connected neural networks. As a starting point, notice that invertible and

differentiable transformations are *composable*; given any two such transformations, their composition is also invertible and differentiable. Furthermore, the determinant of the Jacobian of a composition of functions is given by the product of the determinants of the Jacobians of each function,

$$\det \mathbf{J}[\mathbf{g}_2 \circ \mathbf{g}_1](\mathbf{x}) = \det \mathbf{J}\mathbf{g}_2\left(\mathbf{g}_1(\mathbf{x})\right) \cdot \det \mathbf{J}\mathbf{g}_1(\mathbf{x}). \tag{8}$$

The log-determinant of the full Jacobian for a neural network therefore simply decomposes in a sum of the log-determinants of the Jacobians of each module, $\mathcal{L}_J(\mathbf{x}) = \sum_{k=1}^{L} \log|\det \mathbf{J}\mathbf{g}_k(\mathbf{z}_{k-1})|$. We will focus on the Jacobian term relative to a single submodule $k$ with respect to its input $\mathbf{z}_{k-1}$; with a slight abuse of notation, we will call it $\mathcal{L}_J(\mathbf{z}_{k-1})$. As we remarked, fully connected $\mathbf{g}_k$ are themselves compositions of a linear operation and an element-wise invertible nonlinearity; applying the same reasoning, we then have

$$\mathcal{L}_J(\mathbf{z}_{k-1}) = \sum_{i=1}^{D} \log\left|\sigma'(y_k^i)\right| + \log|\det \mathbf{W}_k| =: \mathcal{L}_J^1(\mathbf{y}_k) + \mathcal{L}_J^2(\mathbf{z}_{k-1}). \tag{9}$$

where $\mathbf{y}_k = \mathbf{W}_k\mathbf{z}_{k-1}$. The first term $\mathcal{L}_J^1$ is a sum of univariate functions of single components of the output of the module, and it can be evaluated easily with few additional operations on top of intermediate outputs of a forward pass; gradients with respect to it can be simply computed via backpropagation, not unlike the $\mathcal{L}_p$ term introduced in section 2.3.

The second term $\mathcal{L}_J^2$ however involves a nonlinear function of the determinant of the weight matrix. From matrix calculus, we know that the derivative is equal to

$$\frac{\partial \log|\det \mathbf{W}_k|}{\partial \mathbf{W}_k} = \left(\mathbf{W}_k^\top\right)^{-1}. \tag{10}$$

Therefore, the computation of the gradient relative to such term involves a matrix inversion, with cubic scaling in the input size.[5] For a fully connected neural network of $L$ layers, given that we have one such operation to perform for each of the layers, the gradient computation for these terms alone would have a complexity of $\mathcal{O}(LD^3)$, thus matching the one which would be obtained if the Jacobian were to be computed via automatic differentiation as discussed in section 2.

It can therefore be seen that these inverses of the weight matrices are the problematic element in the gradient computation. In the next section, we show how this problem can be solved using relative gradients.

## 4 Relative gradient descent for neural networks

We now derive the basic form of the relative gradient, following the approach in [11].[6] The starting point is that the parameters in a neural networks are matrices, in particular invertible in our case. Thus, we can make use of the geometric properties of invertible matrices, while they are usually completely neglected in gradient optimization in neural networks.

**Relative gradient based on multiplicative perturbation** In a classical gradient approach for optimization, we add a small vector $\boldsymbol{\epsilon}$ to a point $\mathbf{x}$ in a Euclidean space. However, with matrices, we are actually perturbing a matrix with another, and this can be done in different ways. In the relative gradient approach, we make a *multiplicative* perturbation of the form

$$\mathbf{W}_k \to (\mathbf{I} + \boldsymbol{\epsilon})\mathbf{W}_k \tag{11}$$

where $\boldsymbol{\epsilon}$ is an infinitesimal matrix. If we consider the effect of such a perturbation on a scalar-valued function $f(\mathbf{W}_k)$, we have

$$f((\mathbf{I} + \boldsymbol{\epsilon})\mathbf{W}_k) - f(\mathbf{W}) = \langle \nabla f(\mathbf{W}_k), \boldsymbol{\epsilon}\mathbf{W}_k \rangle + o(\mathbf{W}_k) = \langle \nabla f(\mathbf{W}_k)\mathbf{W}_k^\top, \boldsymbol{\epsilon} \rangle + o(\mathbf{W}_k) \tag{12}$$

which shows that the direction of steepest descent in this case is given by making $\boldsymbol{\epsilon} = \mu \nabla f(\mathbf{W}_k)\mathbf{W}_k^\top$ where $\mu$ is an infinitesimal step size. Furthermore, when we combine this $\boldsymbol{\epsilon}$ with the definition of a multiplicative update, we find that the best perturbation to $\mathbf{W}$ is actually given as

$$\mathbf{W}_k \to \mathbf{W}_k + \mu \nabla f(\mathbf{W}_k)\mathbf{W}_k^\top \mathbf{W}_k \tag{13}$$

That is, the classical Euclidean gradient is replaced by $\nabla f(\mathbf{W}_k)\mathbf{W}_k^\top\mathbf{W}_k$, i.e. it is multiplied by $\mathbf{W}_k^\top\mathbf{W}_k$ from the right. This is the relative gradient.

A further alternative can be obtained by perturbing the weight matrices from the right, as $\mathbf{W}_k \to \mathbf{W}_k(\mathbf{I} + \boldsymbol{\epsilon})$. A similar derivation shows that in this case, the optimal $\boldsymbol{\epsilon}$ is given by $\mathbf{W}_k\mathbf{W}_k^\top\nabla f(\mathbf{W}_k)$; we refer to this as *transposed relative gradient*. In the context of linear ICA, the properties of the relative and transposed relative gradient were discussed in [49]. This version of the relative gradient might be useful in some cases; for example, the transposed relative gradient can be implemented more straightforwardly in neural network packages where the convention is that vectors are represented as rows.

The relative gradient belongs to the more general class of gradient descent algorithms on Riemannian manifolds [1]. Specifically, relative gradient descent is a first order optimization algorithm on the manifold of invertible $D \times D$ matrices. Almost sure convergence of the parameters to a critical point of the gradient of the cost function can be derived even for its stochastic counterpart, with decreasing step size and under suitable assumptions (see e.g. [8]).

**Jacobian term optimization through the relative gradient**   In section 3, we showed that the difficulty in computing the gradient of the log-determinant is in the terms $\mathcal{L}_J^2$, whose gradient involves a matrix inversion. Now we show that by exploiting the relative gradient, this matrix inversion vanishes. In fact, when multiplying the right hand side of equation (10) by $\mathbf{W}_k^\top\mathbf{W}_k$ from the right we get

$$\left(\mathbf{W}_k^\top\right)^{-1}\mathbf{W}_k^\top\mathbf{W}_k = \mathbf{W}_k \,, \tag{14}$$

and similarly when multiplying by $\mathbf{W}_k\mathbf{W}_k^\top$ from the left. Most notably, we therefore have to perform *no additional operation* to get the relative gradient with respect to this term of the loss; it is, so to say, *implicitly* computed — as we know that the update for the parameters in $\mathbf{W}_k$ with respect to the error term $\mathcal{L}_J^2$ is proportional to $\mathbf{W}_k$ matrix itself.

As for the remaining terms of the loss, $\mathcal{L}_p$ and $\mathcal{L}_J^1$, simple backpropagation allows us to compute the weight updates given by the ordinary gradient in equation (6), which still need to be multiplied by $\mathbf{W}_k^\top\mathbf{W}_k$ to turn it into a relative gradient. We will next see that we can do this avoiding matrix-matrix multiplications, which would be computationally expensive. Note that backpropagation necessarily computes the $\boldsymbol{\delta}_k$ vector in equation (6) and for our model, by applying the relative gradient carefully, we can avoid matrix-matrix multiplication altogether by computing

$$(\Delta\mathbf{W}_k)\,\mathbf{W}_k^\top\mathbf{W}_k \propto \mathbf{z}_{k-1}\left(\left(\boldsymbol{\delta}_k^\top\mathbf{W}_k^\top\right)\mathbf{W}_k\right) \,. \tag{15}$$

Thus, we have a cheap method for computing the gradient of the log-determinant of the Jacobian, and of our original objective function. In appendix D we provide an explanation of how our procedure can be implemented with relative ease on top of existing deep learning packages.

While we so far only discussed update rules for the weight matrices of the neural network, our approach can be extended to include biases. Including bias terms in our multilayer network endows it with stronger approximation capacity. We detail how to do this in appendix F.

**Complexity**  Note that the parentheses in equation (15) stress the point that the relative gradient updates only require matrix-vector or vector-vector multiplications, each of which scales as $\mathcal{O}(D^2)$, in a fixed number at each layer; that is, overall $\mathcal{O}(LD^2)$ operations. They therefore do not increase the complexity of a normal forward pass. Furthermore, the overall complexity with respect to the input size is quadratic, resulting in an overall quadratic scaling with the input size as in normalizing flow methods [44], but without imposing strong restrictions on the Jacobian of the transformation.

**Extension to convolutional layers**  As we remarked in section 2.2, the formalism we introduced includes convolutional neural networks (CNNs) [36]. A natural question is therefore whether our approach can be extended to that case. The first natural question pertains the invertibility of convolutional neural networks; the convolution operation was shown [39] to be invertible under mild conditions (see appendix G), and the standard pooling operation can be by replaced an invertible operation [31]. We therefore believe that the general formalism can be applied to CNNs; this would require the derivation of the relative gradient for tensors. We believe that this should be possible but leave it for future work.

**Invertibility and generation**  Given that invertible and differentiable transformations are composable, as discussed in section 3, invertibility of our learned transformation is guaranteed as long as the

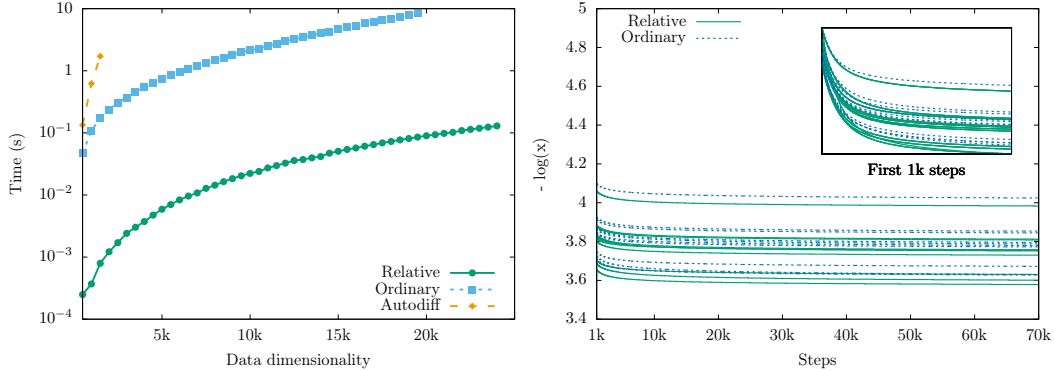

Figure 1: **Left:** Comparison of the average computation times of a single evaluation of the gradient of the log-likelihood; the standard error of the mean is not reported as it is orders of magnitude smaller then the scale of the plot. **Right:** Time-evolution of the negative log-likelihood for deterministic full-batch optimization for the two methods with the same initial points.

weight matrices and the element-wise nonlinearities are invertible. Square and randomly initialized (e.g. with uniform or normally distributed entries) weight matrices are known to be invertible with probability one; invertibility of the weight matrices throughout the training is guaranteed by the fact that the $\mathcal{L}_J^2$ terms would go to minus infinity for singular matrices (though high learning rates and numerical instabilities might compromise it in practice), as in estimation methods for linear ICA [6, 11, 26]. We additionally employ nonlinearities which are invertible by construction; we include more details about this in appendix H. If we are interested in data generation, we also need to invert the learned function. In practice, the cost of inverting each of the matrices is $\mathcal{O}(D^3)$, but the operation needs to be performed only once. As for the nonlinear transformation, the inversion is cheap since we only need to numerically invert a scalar function, for which often a closed form is available.

## 5 Experiments

In the following we experimentally verify the computational advantage of the relative gradient. The code used for our experiments can be found at `https://github.com/fissoreg/relative-gradient-jacobian`.

**Computation of relative vs. ordinary gradient** As a first step, we empirically verify that our proposed procedure using the formulas in section 4 leads to a significant speed-up in computation of the gradient of the Jacobian term. We compare the relative gradient against an explicit computation of the ordinary gradient, as described in section 3, and with a computation based on automatic differentiation, as discussed in section 2.3, where the Jacobian is computed with the JAX package [10]. While the output and asymptotic computational complexity of the ordinary gradient and automatic differentiation methods should be the same, a discrepancy is to be expected at finite dimensionality due to differences in how the computation is implemented. In the experiment, we generate 100 random normally distributed datapoints and vary the dimensionality of the data from 10 to beyond 20,000. We then define a two-layer neural network and evaluate the gradient of the Jacobian. The main comparison is run on a Tesla P100 Nvidia GPU. For the main plots, we deactivated garbage collection. Plots with CPU and further details on garbage collection can be found in appendix H.1. For each dimension we computed 10 iterations with a batch size of 100. Results are shown in figure 1, left. On the y-axis we report the average of the execution times of 100 successive gradient evaluations (forward plus backward pass in the automatic differentiation case). It can be clearly seen that *the relative gradient is much faster*, typically by two orders of magnitude. Autodiff computations could actually only be performed for the smallest dimension due to a memory problem. We report additional details on memory consumption in appendix H.1.

**Optimization by relative vs. ordinary gradient** Since our paper is, to the best of our knowledge, the first one proposing relative gradient optimization for neural networks (though other kinds of natural gradients have been studied [2]), we want to verify that the learning dynamics induced by the

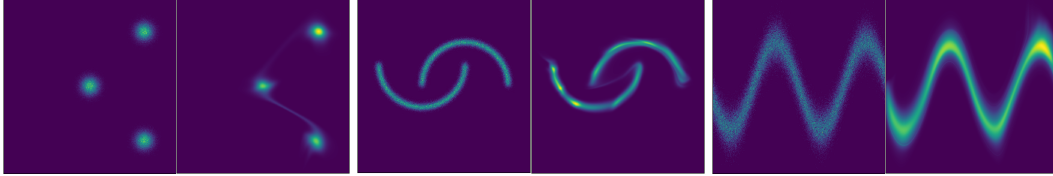

Figure 2: Illustrative examples of 2D density estimation. Samples from the true distribution and predicted densities are shown, in this order, side by side.

relative as opposed to the ordinary gradient do not bias the training procedure towards less optimal solutions or create other problems. We therefore perform a deterministic (full batch) gradient descent for both the relative and the ordinary gradient.[7] We employ 1,000 datapoints of dimensionality 2 and a two-layer neural network. We take 10 initial points and initialize both kinds of gradient descent at those same points. On the x-axis we plot the training epoch, while on the y-axis we plot the value of the loss. Figure 1, right shows the results: there is no big difference between the two gradient methods. There may actually be a slight advantage for the relative gradient, but that is immaterial since our main point here is merely to show that the *relative gradient does not need more iterations* to give the same performance.

Combining these two results, we see that the proposed relative gradient approach leads to a *much faster optimization* than the ordinary gradient. Perhaps surprisingly, the results exhibit a rather constant speed-up factor of the order of 100 although the theory says it should be changing with the dimension $D$; in any case, the difference is very significant in practice.

**Density estimation** Although our main contribution is the computational speed-up of the gradient computation demonstrated above, we further show some simple results on density estimation to highlight the potential of the relative gradient used in conjuction with the unconstrained factorial approximation in section 2.1. We use a fairly simple feedforward neural network with a smooth version of leaky-ReLU as activation function. Our empirical results show that this system, despite having quite *minimal fine-tuning* (details in appendix H.3), *achieves competitive results on all the considered datasets* compared with existing models—which are all tailored and fine-tuned for density estimation. First, we show in Figure 2 different toy examples that showcase the ability of our method to convincingly model arbitrarily complex densities. Second, in order to show the viability of our method in comparison with well-established methods we perform, as in [43], unconditional density estimation on four different UCI datasets [16] and a dataset of natural image patches (BSDS300) [41], as well as on MNIST [37]. The results are shown in Table 1. To achieve a fair comparison across models, the number of parameters was tuned so that the number of trainable parameters are as similar as possible. Note that, as we can perform every computation efficiently, all the experiments are suitable to run on usual hardware, thus avoiding the need of hardware accelerators such as GPUs. As a final remark, the reported results make no use of batch normalization, dropout, or learning-rate scheduling. Therefore, it is sensible to expect even better results by including them in future work.

Table 1: Test log-likelihoods (higher is better) on unconditional density estimation for different datasets and models (same as in Table 1 of [43]). Models use a similar number of parameters; results show mean and two standard deviations. Best performing models are in bold. More details in appendix H.3

|  | POWER | GAS | HEPMASS | MINIBOONE | BSDS300 | MNIST |
|---|---|---|---|---|---|---|
| Ours | $0.065 \pm 0.013$ | $6.978 \pm 0.020$ | $-21.958 \pm 0.019$ | $-13.372 \pm 0.450$ | $151.12 \pm 0.28$ | $-1375.2 \pm 1.4$ |
| MADE | $-3.097 \pm 0.030$ | $3.306 \pm 0.039$ | $-21.804 \pm 0.020$ | $-15.635 \pm 0.498$ | $146.37 \pm 0.28$ | $-1380.8 \pm 4.8$ |
| MADE MoG | $\mathbf{0.375 \pm 0.013}$ | $7.803 \pm 0.022$ | $\mathbf{-18.368 \pm 0.019}$ | $-12.740 \pm 0.439$ | $150.84 \pm 0.27$ | $\mathbf{-1038.5 \pm 1.8}$ |
| Real NVP (10) | $0.182 \pm 0.014$ | $\mathbf{8.357 \pm 0.019}$ | $-18.938 \pm 0.021$ | $\mathbf{-11.795 \pm 0.453}$ | $\mathbf{153.28 \pm 1.78}$ | $-1370.7 \pm 10.1$ |
| Real NVP (5) | $-0.459 \pm 0.010$ | $6.656 \pm 0.020$ | $-20.037 \pm 0.020$ | $-12.418 \pm 0.456$ | $151.76 \pm 0.27$ | $-1323.2 \pm 6.6$ |
| MAF (5) | $-0.458 \pm 0.016$ | $7.042 \pm 0.024$ | $-19.400 \pm 0.020$ | $-11.816 \pm 0.444$ | $149.22 \pm 0.28$ | $-1300.5 \pm 1.7$ |
| MAF (10) | $-0.376 \pm 0.017$ | $7.549 \pm 0.020$ | $-25.701 \pm 0.025$ | $-11.892 \pm 0.459$ | $150.46 \pm 0.28$ | $-1313.1 \pm 2.0$ |
| MAF MoG (5) | $0.192 \pm 0.014$ | $7.183 \pm 0.020$ | $-22.747 \pm 0.017$ | $-11.995 \pm 0.462$ | $152.58 \pm 0.66$ | $-1100.3 \pm 1.6$ |

# 6 Conclusions

Using relative gradients, we proposed a new method for exact optimization of objective functions involving the log-determinant of the Jacobian of a neural network, as typically found in density estimation, nonlinear ICA, and related tasks. This allows for employing models which, unlike typical alternatives in the normalizing flows literature, have no strong limitation on the structure of the Jacobian. We use modules with fully connected layers, thus strictly generalizing normalizing flows with triangular Jacobians, while still supporting efficient combination of forward and backward pass. These neural networks can represent a larger function class than autoregressive flows, which, despite being universal approximators for density functions, can only represent transformations with triangular Jacobians. Our method can therefore provide an alternative in settings where more expressiveness is needed to learn a proper inverse transformation, such as in identifiable nonlinear ICA models.

The relative gradient approach proposed here is quite simple, yet rather powerful. The importance of the optimization of the log-determinant of the Jacobian is well-known, but it has not been previously shown that there is a way around its difficulty without restricting expressivity. Now that we have shown that the optimization of this term can be done quite cheaply, a substantial fraction of the research in the field can be reformulated in stronger terms and with more generality.

## Broader impact

As this paper presents novel theoretical results in unsupervised learning, the authors do not see any immediate ethical or societal concern. An important aspect of our paper is the improvement in computational efficiency with respect to naive methods. This can hopefully lead to reduced energy consumption to achieve comparable model performance.

## Acknowledgments

A.H. was supported by a Fellowship from CIFAR, and by the DATAIA convergence institute as part of the "Programme d'Investissement d'Avenir", (ANR-17-CONV-0003) operated by Inria.
L.G. started working on this project while on an ELLIS exchange, hosted by the Parietal team at Inria, Saclay.
We thank Vincent Stimper, Patrick Putzky, Cyril Furtlehner and Ilyes Khemakhem for valuable comments on an earlier draft of this paper, and Isabel Valera and Guillaume Charpiat for helpful comments and tips. L.G. additionally thanks Roma Beaufret and Alexis Bozio for helpful support.

## Footnotes

[2]The forward transformation could also be parameterized, but here we only explicitly parameterize its inverse.

[3]Note that invertible neural networks provide the possibility to not save, but rather recompute the intermediate activations during the backward pass, thus providing a memory efficient approach to backpropagation [18].

[4]Comprehensive reviews on normalizing flows can be found in [42, 35]. Other related methods are reviewed in appendix B.

[5]Though slightly more favorable exponents can in principle be obtained, see appendix C.

[6]For linear blind source separation, this approach also corresponds to the natural gradient, which can be justified with an information-geometric approach [2].

[7]Notice that there's no need to compare to autodiff in this case because the computed gradient should be exactly the same as the ordinary gradient with the formulas in section 3.

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
