[Supplementary Material]

# APPENDIX

## A  Backpropagation in neural networks

We will follow [46], Chapter 7, section 7.3.3 for the notation. Let us define a two-layer neural network

$$\mathbf{g}_\theta(\mathbf{x}) = \boldsymbol{\sigma}\left(\mathbf{W}_2\boldsymbol{\sigma}\left(\mathbf{W}_1\mathbf{x}\right)\right) \tag{16}$$

where we also define

$$\mathbf{z}_2 = \boldsymbol{\sigma}\left(\mathbf{W}_2\mathbf{z}_1\right)$$
$$\mathbf{z}_1 = \boldsymbol{\sigma}\left(\mathbf{W}_1\mathbf{x}\right).$$

and

$$\mathbf{u}_2 = \boldsymbol{\sigma}'\left(\mathbf{W}_2\mathbf{z}_1\right)$$
$$\mathbf{u}_1 = \boldsymbol{\sigma}'(\mathbf{W}_1\mathbf{x})$$

and

$$\mathbf{y}_2 = \mathbf{W}_2\mathbf{z}_1$$
$$\mathbf{y}_1 = \mathbf{W}_1\mathbf{x}$$

We need to consider the contributions to the objective function due to the terms $\mathcal{L}_p$ and $\mathcal{L}_J^1$ (the contribution due to $\mathcal{L}_J^2$ will be dealt with separately). For $\mathcal{L}_p$, we define

$$e(x) = \frac{\partial}{\partial x}\log p(x')|_{x'=x}$$

and

$$\mathbf{e} = \begin{pmatrix} e(z_2^1) \\ e(z_2^2) \\ \vdots \\ e(z_2^D) \end{pmatrix}$$

To deal with the terms in $\mathcal{L}_J^1$, we define

$$h(x) = \frac{\partial}{\partial x}\log x'|_{x'=x} \tag{17}$$

$$= \frac{1}{x} \tag{18}$$

and

$$\mathbf{h}_k = \begin{pmatrix} h(u_k^1) \\ h(u_k^2) \\ \vdots \\ h(u_k^D) \end{pmatrix}$$

for $k = 1, 2$. During forward propagation, we store the $\mathbf{D}_k = \mathrm{diag}\left(\boldsymbol{\sigma}'\left(\mathbf{y}_k\right)\right)$ for $k = 1, 2$,

$$\mathbf{D}_k = \begin{pmatrix} \sigma'(y_k^1) & 0 & \cdots & 0 \\ 0 & \sigma'(y_k^2) & \cdots & 0 \\ \vdots & \vdots & \ddots & \vdots \\ 0 & 0 & \cdots & \sigma'(y_k^D) \end{pmatrix}$$

and the $\mathbf{G}_k = \mathrm{diag}\left(\boldsymbol{\sigma}''\left(\mathbf{y}_k\right)\right)$ for $k = 1, 2$,

$$\mathbf{G}_k = \begin{pmatrix} \sigma''(y_k^1) & 0 & \cdots & 0 \\ 0 & \sigma''(y_k^2) & \cdots & 0 \\ \vdots & \vdots & \ddots & \vdots \\ 0 & 0 & \cdots & \sigma''(y_k^D) \end{pmatrix}$$

for example, if the nonlinearity were a sigmoid function $\sigma(x) = (1 + e^{-x})^{-1}$, the second derivative would be $\sigma''(x) = \sigma(x)(1 - \sigma(x))\,(1 - 2\sigma(x))$. Then

$$\boldsymbol{\delta}_2 = \mathbf{D}_2 \mathbf{e} + \mathbf{G}_2 \mathbf{h}_2$$

and

$$\boldsymbol{\delta}_1 = \mathbf{D}_1 \mathbf{W}_2 \boldsymbol{\delta}_2 + \mathbf{G}_1 \mathbf{h}_1$$

In general, the following recursive relationship holds

$$\boldsymbol{\delta}_k = \mathbf{D}_k \mathbf{W}_{k+1} \boldsymbol{\delta}_{k+1} + \mathbf{G}_k \mathbf{h}_k \tag{19}$$

Which results in the update rule

$$\Delta \mathbf{W}_k = -\mu \mathbf{z}_{k-1} \boldsymbol{\delta}_k^\top \,,$$

where $\mathbf{z}_0 = \mathbf{x}$. Notice that the only necessary operations are vector-matrix, matrix-vector and vector-vector multiplications.

## A.1 Relative gradient

Now if we want to use the relative/natural gradient trick each of these terms needs to be multiplied by $\mathbf{W}_k^\top \mathbf{W}_k$ from the right.

$$\Delta \mathbf{W}_k = -\mu \mathbf{z}_{k-1} \boldsymbol{\delta}_k^\top \mathbf{W}_k^\top \mathbf{W}_k \,.$$

**Terms in $\mathcal{L}_J^2$** The terms in $\mathcal{L}_J^2$, consisting of $\log |\mathbf{W}_k|$ give as gradient $\left(\mathbf{W}_k^\top\right)^{-1}$. This requires a $D \times D$ matrix inversion for each of the matrices. Our strategy to avoid it is to substitute the ordinary gradient with a relative gradient, where we multiply the gradient (with respect to the whole objective but for each layer separately) by $\mathbf{W}_k^\top \mathbf{W}_k$ from the right. In this case, the updates for the $\mathbf{W}_k$ terms simply become proportional to the $\mathbf{W}_k$ themselves. Therefore, the update rule becomes

$$\Delta \mathbf{W}_k = -\mu (\mathbf{z}_{k-1} \boldsymbol{\delta}_k^\top \mathbf{W}_k^\top \mathbf{W}_k + \mathbf{W}_k) \,. \tag{20}$$

As we already noted, the operations involved in these updates can be performed in a way such that no matrix-matrix multiplication needs to be performed – only matrix-vector and vector-vector multiplication. This is more apparent when the update rules are rewritten as below

$$\Delta \mathbf{W}_k = -\mu \left( \mathbf{z}_{k-1} \left( \left(\boldsymbol{\delta}_k^\top \mathbf{W}_k^\top\right) \mathbf{W}_k \right) + \mathbf{W}_k \right) \,. \tag{21}$$

# B Related work

In the following, we present a review of related work in tractable deep density estimation and invertible neural networks.

**Normalizing flows** The modern conception of normalizing flows was introduced in [50], which discussed density estimation through the composition of simple maps. In [45], it was then proposed that deep density models implemented through neural networks could be used in order to construct bijective maps to a representation space and obtain normalized probability density estimates. Since then, the focus mainly shifted to scalability; [14, 15] introduced scalable architectures, further refined in [33] to make them more scalable and suitable for practical applications; [44] applied the results to variational inference. Comprehensive reviews on normalizing flows can be found in [42, 35].

**Autoregressive flows** Autoregressive flows are among the most used in practice. They involve maps which can be written as $z_i' = \tau(z_i; \boldsymbol{h}_i)$, with $\boldsymbol{h}_i = c_i (\mathbf{z}_{<i})$. $\tau$ is termed the *transformer* and is a strictly monotonic function of $z_i$, and $c_i$ is termed the $i$-th *conditioner*. Its constraint is that the $i-$th conditioner can only take variables with dimension indices less than $i$ as an input. This results in an overall transformation with a triangular Jacobian; the determinant is therefore tractable and can be computed in $\mathcal{O}(D)$ time. Autoregressive flows differ in the way the transformer and conditioner are implemented; most commonly used are affine autoregressive flows [14, 15, 34, 43, 33] and non-affine neural transformers [25].

**Linear flows** A strict generalization of autoregressive flows, where the Jacobian is not constrained to be triangular, is given by linear flows, which are essentially transformations of the form $\mathbf{z}' = \mathbf{Wz}$, where $\mathbf{W}$ is a $D \times D$ invertible matrix. The Jacobian of the trasformation is simply $\mathbf{W}$ and both

computing and optimizing its determinant takes time $O(D^3)$ in general. To obtain a better scaling behaviour, [14] and [22] proposed to parameterize the invertible $\mathbf{W}$ matrix via matrix decomposition. One possibility is to compute the $\mathbf{PLU}$ decomposition of $\mathbf{W}$ and optimize the $\mathbf{L}$ and $\mathbf{U}$ triangular transformations. The drawback in this approach is that the permutation matrix $\mathbf{P}$ cannot be learned. A more flexible alternative is to consider the $\mathbf{QR}$ decomposition of $\mathbf{W}$, where $\mathbf{Q}$ is an orthogonal matrix and $\mathbf{R}$ is upper triangular. However computing $\mathbf{Q}$ in full generality requires $O(D^3)$ operations, matching the complexity of the naive optimization of linear flows. [52] showed that we can apply the $\mathbf{Q}$ transformation as a sequence of at most $D$ symmetry transformations each taking linear time, effectively making it possible to compute and optimize the $\mathbf{QR}$ parameterization of $\mathbf{W}$ in $O(D^2)$ time; note however that the sequential nature of the computation makes the method unsuitable for optimization on hardware accelerators. An experimental comparison of the performance of the $\mathbf{PLU}$ and $\mathbf{QR}$ decompositions against the direct optimization of $\mathbf{W}$ is found in [22].

**Flows based on residual transformations** Another class of normalizing flows is based on invertible transformations of the form $\mathbf{z}' = \mathbf{z} + g_\phi(\mathbf{z})$; this kind of flows are termed *residual flows*. Two main approaches can be applied to build invertible residual flows: the first exploits the matrix determinant lemma and also results in determinants with $\mathcal{O}(D)$ computation time; however, there is no analytical way of computing their inverse. Examples of these approaches are Sylvester flows [53], planar flows [44] and radial flows [50, 44]. The second approach is that of contractive flows [5]: in this case, the determinant can not be computed simply; likelihood-based training of these models therefore needs to rely on a Hutchkinson's trace based approximation to the exact log-likelihood.

**Continuous time flows** A separate line of work focuses on building *continuous flows*; in these approaches, the flow's infinitesimal dynamics is parametrized in continuous time, and the corresponding transformation is then found by integration [13, 19]; Hamiltonian Flows [44] can also be regarded as such kind of flows.

**Other works** Recently, many works have proposed ways of incorporating convolutional modules in normalizing flows, for example see [33, 22, 32]. In particular, [17] presents a formalization of the problem which bears some similarities to ours, while focusing on convolutional layers instead of fully connected ones. Other work has been dedicated to constructing invertible neural networks, see for example [3, 31, 18].

# C   Complexity of mathematical operations involved in gradient computation

We want to characterize the complexity of computing

$$\nabla_{\boldsymbol{\theta}} \log |\det \mathbf{Jg}_{\boldsymbol{\theta}}(\mathbf{x})|, \tag{22}$$

where $\mathbf{g}_{\boldsymbol{\theta}}$ is a neural network.

We will first recapitulate the computational complexity of the main mathematical operations we employ (see e.g. [55]). Then we'll recapitulate the complexity of forward evaluation and backpropagation in neural networks. Finally, we'll discuss the implications on the complexity of computing the term in equation (22) with the three methods discussed in the paper — namely, based on automatic differentiation, the standard computation described in section 3 and the relative gradient based computation.

## C.1   Matrix operations

**Matrix-vector and vector-vector multiplication**   The multiplication of a $D \times D$ matrix with a $D \times 1$ vector scales as $\mathcal{O}(D^2)$. Same for the outer product between two vectors of dimension $D \times 1$.

**Matrix-matrix multiplication**   For the multiplication of two square matrices of size $D \times D$

- An implementation of the Bareiss algorithm would scale as $\mathcal{O}(D^3)$;
- An implementation of the Strassen algorithm would scale as $\mathcal{O}(D^{2.807\cdots})$ ;
- An implementation of the Coppersmith-Winograd algorithm would scale as $\mathcal{O}(D^{2.373\cdots})$ .

In practice, what is usually implemented in linear algebra libraries is some flavor of the Strassen algorithm (this is because the Coppersmith-Winograd algorithm, while having a more favorable asymptotic behaviour, is effectively slower if $D$ is not extremely high).

**Matrix inversion**   To find the inverse of a matrix of size $D \times D$

- An implementation of Gauss-Jordan elimination would scale as $\mathcal{O}(D^3)$;
- An implementation of the Strassen algorithm would scale as $\mathcal{O}(D^{2.807\cdots})$ ;
- An implementation of the Coppersmith-Winograd algorithm would scale as $\mathcal{O}(D^{2.373\cdots})$ .

**Determinant**   To find the determinant of a matrix of size $D \times D$

- An implementation of the Bareiss algorithm would scale as $\mathcal{O}(D^3)$;
- Algorithms based on fast matrix multiplication scale as $\mathcal{O}(D^{2.373\cdots})$ .

For simplicity, in most of our considerations on complexity we assume that the computation of the determinant, the computation of the inverse and the multiplication of two square matrices have cubic cost. Notice that the cost of these operations always dominates over that of matrix-vector and vector-vector multiplication.

## C.2   Other operations involved in the Jacobian term computation

Other operations turn out to be ininfluent on the overall computational complexity. Namely logarithms, absolute values, sums have no relevant effect in terms of asymptotic scaling, since their computational cost is dominated by that of the most expensive matrix operations listed above.

## C.3   Complexity of neural network operations

**Forward pass in a neural network**   The complexity of the forward pass in a neural network depends on the neural network structure. For simplicity, we will consider fully connected Neural Networks, which due to their dense structure will provide an upper bound for the complexity of most of the nets used in practice. Given an input vector, the forward pass is comprised of a sequential series of matrix-vector operations, plus elementwise operations on the resulting vector. The matrix-vector operations dominate the complexity; for an $L$ layer neural network, there are $L$ such operations. Therefore, for data of dimensionality $D$, the complexity of a forward pass in a Neural Network for a single data sample is $\mathcal{O}(LD^2)$.

**Minibatching**   The objectives should, in principle, be optimized on the full batch. Stochastic optimization [9] relies on the idea that the update steps in the optimization process can be performed on subsets of the whole training data, called minibatches. In practice these objectives will always be computed on minibatches, so the expected value must be substituted with its empirical estimate over a single minibatch. The minibatch size should in principle be considered when considering how the algorithm scales. In the remainder, however, we will neglect this term, as minibatches used in practice are usually quite small.

**Gradient computation**   On top of this, we also need to consider the gradient computation. Since the gradient is taken over the scalar loss function, this implies (through backpropagation or reverse mode differentiation) no increase in the asymptotic computational cost. We further elaborate on this in the next section.

## C.4   Computing the Jacobian with automatic differentiation

**Jacobian through automatic differentiation**   Automatic differentiation [4] includes two main operational modes: the forward mode and the backward mode. Consider the computation of the Jacobian of a function $\mathbf{g}_\theta) : \mathbb{R}^D \to \mathbb{R}^d$. The complexity of computing the Jacobian will depend on whether we use forward or reverse mode AD. This changes the complexity of the operation:

- forward mode requires $D\,c\,\mathrm{ops}(\mathbf{g}_\theta)$ operations, where $D$ is the dimensionality of the data and $c$ is a constant, $c < 6$ and typically $c \in [2,3]$ (see [21]);
- reverse mode requires $d\,c\,\mathrm{ops}(\mathbf{g}_\theta)$ operations.

In the case of dimensionality reduction, reverse mode differentiation (of which backpropagation represents an instance) is clearly more efficient. This is the case when the output of the function is scalar ($d = 1$); thus, this explains our claim that gradients computation with backpropagation implies no increase in the asymptotic computational cost with respect to the forward pass alone.

For neural networks where all layers (including input and output) have the same size, both methods result in the same complexity. So in that case neither is better in terms of computational complexity — though in practice it is known that reverse mode performs better [40]. For such neural networks (including those we consider) therefore, given that $\text{ops}(\mathbf{g}_\theta)$ is $\mathcal{O}(LD^2)$, the overall complexity of the Jacobian computation via automatic differentiation is $\mathcal{O}(LD^3)$.

The gradient of the objective can then be computed via backpropagation; however, the forward evaluation is what dominates the overall complexity.

**Standard and relative gradient computations**   The evaluation of the two terms $\mathcal{L}_p$ and $\mathcal{L}_J^1$ requires a forward pass of the neural networks, thus scaling as $\mathcal{O}(LD^2)$. As we discussed, backpropagation to compute the gradient does not increase the overall cost. For $\mathcal{L}_J^2$, as we have shown, the gradient can be computed without need to actually evaluate the corresponding term (that is, side-stepping the determinant computation). However, the standard computation of the gradient still requires computing inverses of all the weight matrices, resulting in a cubic cost operation for each layer — thus utimately in $\mathcal{O}(LD^3)$ cost.

When using the relative gradient, this inversion can be avoided, and computing the gradients of $\mathcal{L}_J^2$ implies *no additional costs*. The overall cost of the gradient computation is therefore simply $\mathcal{O}(LD^2)$.

## D   Implementation details

To efficiently optimize our objective (e.g. equation (3) in the main paper) we need to implement a variant of the backpropagation algorithm as detailed in appendix A. In particular, we need to compute the updates (equation (15) in the main paper) avoiding expensive matrix-matrix multiplications. This section is devoted to the description of an implementation strategy that takes advantage of Automatic Differentiation (AD), in order to have full flexibility in the definition of our model architectures and loss functions.

Although all modern deep learning frameworks include automatic differentiation libraries, they implement the standard backpropagation algorithm. To implement our variant, we have two straightforward alternatives:

- tweak some existing AD libraries to let us access the extra terms we need;
- implement our own AD library with the extra functionality we need.

The second alternative is easily excluded as we don't want to reinvent the wheel and the development effort would be too much. The first alternative is somewhat viable, but not future proof; we would be faced with the need to support our own modifications on top of the AD library we use.

We obviate to these problems with a little trick: we introduce in our architectures some dummy layers to accumulate the partial results that the standard backpropagation computes in the backward pass. This approach solves the previous problems by being:

- universal: it can be easily implemented on top of whatever AD library that computes reverse-mode AD, without tweaking the internals of the library;
- efficient: the dummy layer operations are $\mathcal{O}(1)$.

### D.1   The Accumulator layer

To obtain the gradient updates (20) we need to compute the $\delta$ terms (19). To better understand what these terms represent, we can consider a simple 2-layers "scalar" network, i.e. a network in which inputs, outputs and weights are scalar values:

$$f(x; \boldsymbol{w}) = w_2 \sigma(w_1 x) \tag{23}$$
$$= w_2 \sigma(y_1)$$
$$= w_2 z_1$$
$$= y_2$$

where $\boldsymbol{w}$ is the vector of scalar parameters, $\sigma$ is the activation function of choice and

$$y_1 = w_1 x, \quad y_2 = w_2 z_1, \quad z_1 = \sigma(y_1).$$

Given a loss function $\mathcal{L}$, the gradient of $\mathcal{L}$ with respect to $w_1$ is easily computed with application of the chain rule

$$\frac{\partial \mathcal{L}}{\partial w_1} = \frac{\partial \mathcal{L}}{\partial y_2} \frac{\partial y_2}{\partial z_1} \frac{\partial z_1}{\partial y_1} \frac{\partial y_1}{\partial w_1} \tag{24}$$

In this simple case, it is easy to isolate $\delta$ in the gradient equation:

$$\frac{\partial \mathcal{L}}{\partial w_1} = \delta_1 \frac{\partial y_1}{\partial w_1} \tag{25}$$

Reverse mode AD libraries necessarily compute all the partial derivatives in (24) and thus the $\delta_1$ term we need. Unfortunately, the partial results are usually not accessible by the users. To access such terms without dealing with the internals of the AD libraries, we can introduce a parameterized function

$$a(x; \lambda) = x + \lambda$$

and redefine our scalar network as

$$f(x; \boldsymbol{w}) = w_2 \sigma(a(y_1)) \tag{26}$$

The gradient with respect to $w_1$ becomes

$$\frac{\partial \mathcal{L}}{\partial w_1} = \frac{\partial \mathcal{L}}{\partial y_2} \frac{\partial y_2}{\partial z_1} \frac{\partial z_1}{\partial a} \frac{\partial a}{\partial y_1} \frac{\partial y_1}{\partial w_1} \tag{27}$$

The introduction of $a$ is only a trick; in order not to modify the gradients nor the behaviour of the scalar network, we require

$$a(y_1) = y_1 \tag{28}$$
$$\frac{\partial z_1}{\partial a} = \frac{\partial z_1}{\partial y_1}$$
$$\frac{\partial a}{\partial y_1} = 1$$

which is easily achieved by setting $\lambda = 0$.

The benefit of introducing this accumulator layer $a$ is that now we can ask the AD library to compute the gradients with respect to the dummy parameter $\lambda$; it is easy to verify that

$$\frac{\partial a}{\partial \lambda} = \delta_1 \tag{29}$$

thus making it possible to obtain the $\delta$ terms we need to compute (20).

# E  Universal approximation capacity in normalizing flows

Universal approximation for densities is a property often discussed in the context of autoregressive normalizing flows. It can be shown, based on the proof of existence and non-uniqueness of solutions to the nonlinear ICA problem [29], that any distribution can be mapped onto a factorized base distribution by an invertible function with triangular Jacobian, provided that the function class used for this mapping is large enough. Normalizing flows with triangular Jacobians and a high number of parameters therefore have this approximation capacity (see e.g. [25]). However, they can obviously not represent all possible *functions* — but only those with triangular Jacobians. They can therefore not be used to learn proper inverse functions and perform useful feature extraction.

A more general notion of universal approximation is the one usually discussed in the neural network literature, that is — universal approximation for functions. It has been shown that standard multilayer feedforward networks can approximate any continuous function to any degree of accuracy. For example, [38] proved that a standard multilayer feedforward network with a locally bounded piecewise continuous activation function can approximate any continuous function to any degree of accuracy if and only if the network's activation function is not a polynomial. Biases also play a crucial role in this proof, as universal approximation capacity wouldn't be possible without.

While the proof above does not directly apply to our case, since it requires hidden layers with arbitrary width, we discuss how to incorporate biases in our training procedure in appendix F, in order to increase the expressivity of our model. We describe the nonlinearities we employed in appendix H.

# F  Relative gradient for the augmented matrix

In order to allow for the training of neural networks with biases, we present a heuristic based on the fact that affine transformations involving vector-matrix products plus biases can be represented as a single matrix operation through the formalism of the augmented matrix (see e.g. [46]).

Linear affine operations of the form $\mathbf{y} = \mathbf{W}\mathbf{x} + \mathbf{b}$ can be represented via an augmented matrix as follows

$$\begin{bmatrix} \mathbf{y} \\ 1 \end{bmatrix} = \left[ \begin{array}{cc|c} & \mathbf{W} & \mathbf{b} \\ 0 & \dots & 0 & 1 \end{array} \right] \begin{bmatrix} \mathbf{x} \\ 1 \end{bmatrix} = \overline{\mathbf{W}} \begin{bmatrix} \mathbf{x} \\ 1 \end{bmatrix}, \tag{30}$$

where we refer to the matrix $\overline{\mathbf{W}}$ as *augmented matrix*.

The question is whether the relative gradient trick can be applied to the augmented matrix. The main issue is that we would like, throughout our optimization procedure, to remain on the manifold of augmented matrices; that is, we do not want to change the last row of $\overline{\mathbf{W}}_k$. Therefore, the problem becomes a constrained optimization problem.

**The $\mathcal{L}_J^2$ term** It is easy to see that $\det \overline{\mathbf{W}}_k = \det \mathbf{W}_k$. The ordinary gradient for all terms in the last column and row of $\overline{\mathbf{W}}_k$ will therefore be equal to zero, and this will not be changed by the relative gradient trick; therefore, the contribution of this term will not lead us away from the manifold of augmented matrices.

**The $\mathcal{L}_p$ and $\mathcal{L}_J^1$ terms** Both the $\mathbf{y}_k$ and $\mathbf{z}_k$ terms will however be influenced by the presence of biases, so the gradients on the first $D$ elements of the last column (that is $\mathbf{b}_k$) will be nonzero. Through the multiplication with $\overline{\mathbf{W}}_k^\top \overline{\mathbf{W}}_k$, the updates given by the relative gradient on the last row of $\overline{\mathbf{W}}_k$ will therefore in general be nonzero, thus implying moving outside of the manifold we are interested in.

To solve this issue, we use a projected gradient algorithm, enforcing that the update for the last row of $\overline{\mathbf{W}}_k$ at each step is equal to zero. We call this algorithm *projected relative gradient descent*.

In practice, we can use the augmented matrix formalism to apply the relative trick to the full parameters and then extract only the updates for the parameters of interest $\mathbf{W}$, $\mathbf{b}$ disregarding the dummy row in (30). Denoting by $\mathbf{G}$ the gradients of $\mathbf{W}$ and by $\mathbf{g}_b$ the gradients of $\mathbf{b}$, we can compute the relative gradients as

$$\left[ \begin{array}{c|c} \mathbf{G} & \mathbf{g}_b \\ \hline \mathbf{g} & g \end{array} \right] \overline{\mathbf{W}}^\top \overline{\mathbf{W}} = \left[ \begin{array}{c|c} \mathbf{G}\mathbf{W}^\top\mathbf{W} + \mathbf{g}_b\mathbf{b}^\top\mathbf{W} & \mathbf{G}\mathbf{W}^\top\mathbf{b} + \mathbf{g}_b\mathbf{b}^\top\mathbf{b} + \mathbf{g}_b \\ \hline \dots & \dots \end{array} \right] \tag{31}$$

The relative gradient updates we need are then given by

$$\Delta \mathbf{W} \rightarrow \mathbf{G} \mathbf{W}^\top \mathbf{W} + \mathbf{g}_b \left( \mathbf{b}^\top \mathbf{W} \right) \tag{32}$$

$$\Delta \mathbf{b} \rightarrow \mathbf{G} \left( \mathbf{W}^\top \mathbf{b} \right) + \mathbf{g}_b (1 + \mathbf{b}^\top \mathbf{b}) \tag{33}$$

Note that $\mathbf{G}$ is nothing more then the standard backpropagation update (6), thus we can efficiently compute $\Delta \mathbf{W}$ by avoiding matrix-matrix multiplications as in (15). For $\Delta \mathbf{b}$ we can directly avoid matrix-matrix multiplications by taking some care in the evaluation of (33).

## G   Convolutions

The convolutional neural network [56] is composed of modules whose main components are: (i) a convolution layer; (ii) a pooling layer; (iii) a nonlinearity.

**The convolution operation**   We follow the same notation as in [56]. Typically, inputs to the convolution layers are order 3 tensors with size $H^l \times W^l \times D^l$. A convolution kernel is also an order 3 tensor with size $H \times W^l \times D^l$. If $D$ convolutions are used, this results in a order 4 tensor $\mathbb{R}^{H \times W^l \times D^l \times D}$ of parameters. If the input is $H \times W^l \times D^l$ and the kernel size is $H \times W^l \times D^l \times D$, the convolution result has size $(H^l - H + 1) \times (W^l - W + 1) \times D$. In our setting, note that the number of channels which can be used in practice is constrained, due to the formula in equation (3), which requires the input and output dimensionalities to be equal.

**Are convolutional neural networks invertible?**   The convolution operation was shown to be invertible under some mild conditions. See [39] and [17], section 3.3, describing how Gaussian (or Uniform) sampled $c \times c \times r \times r$ parameter tensors will yield invertible convolutional layers with probability 1.

The pooling layer can be substituted with an invertible counterpart (see [31], section 3; or [17], figure 3), which basically becomes a tensorial extension of the permutation operation. As usual, an invertible nonlinearity can be chosen.

**Relative gradient for the convolution**   For a convolution layer that preserves the number of channels in the input, we can directly write the operation in the form of a square matrix. In this case we can compute the relative gradient as explained in section 4, and we can obtain the gradients with respect to the filter entries by careful application of the chain rule. We however leave the precise theoretical derivation and experiments for future work.

Figure 3: Comparison of the average computation times of a single evaluation of the gradient of the log-likelihood over a batch of size 100. Values are the mean over 5 steps, and the experiments have been run 5 times on a CPU cluster.

## H Experiments

### H.1 Computation of relative vs. ordinary gradient

**Computational cost** In section 5 and figure 1 we compared the computational cost of computing log-likelihood gradients with our newly proposed method and a naive backpropagation implementation when using hardware accelerators. Specifically, we used one Tesla P100 GPU card equipped with 16 GB of dedicated memory and circa 3500 computing cores. In figure 3 we show the same comparison for a computation platform comprising 48 cpu threads (Intel Xeon Processor E5-2650 v4 @ 2.20 GHz base frequency, 2.90 GHz max frequency) operating in parallel with about 250 GB of available RAM memory. It is hard to spot the expected theoretical improvement from $O(D^3)$ to $O(D^2)$, but a practical gain of about 2 orders of magnitude in computation time emerges in favor of the relative gradient computation.

In order to directly compare the execution times disregarding bottlenecks due to memory operations, we performed all of the experiments with no garbage collection. Anyways, by using always the same batch we made our experiments not very memory intensive and repeating the experiments with garbage collection enabled didn't show any substantial difference; we therefore don't report the plot.

**Memory consumption** It is usual in deep learning to be constrained by the memory consumption of the models in use, as the available memory on hardware accelerators is typically scarce. To operate, a network needs to store the data, the intermediate activations (needed to compute gradients) and the parameters. For our simple architecture, the bottleneck is the storage of the parameters; this is because we don't employ very deep architectures, so the amount of intermediate activations to store is limited, and the size of the parameters grows quadratically with respect to the data size, meaning that parameters storage clearly dominate over data storage (this is assuming that data are loaded in small minibatches, which is the norm). This is certainly problematic for very high-dimensional datasets (i.e. high definition images) but even from this point of view we have a clear advantage over an explicit optimization of the Jacobian term with automatic differentiation. In this latter case, in fact, we need to compute the full Jacobian of the affine transformations for each individual data point; like for the weight matrices, the size of these terms grows quadratically with the input size, further increasing the memory footprint of the optimization procedure.

Figure 4: Comparison of the memory consumption for a single gradient evaluation. With D = 5000 our simplified analysis predicts a lower bound in the memory consumption of 400 MB for storing the parameters and the computed gradients; given that at startup time we observe a base memory consumption of almost 200 MB (computing environment + loaded libraries) we can see that our relative gradient implementation comes very close to the theoretical limit. For the naive autodiff implementation, instead, we compute a lower bound of 10.4 GB, which is approximately reflected in the empirical measurements (maximum consumption is almost 13 GB). Note: memory consumption for the autodiff case is reported in GiB, effectively making the scale of the plot one order of magnitude higher then in the relative gradient plot.

As a simple example, we can compare the approximate memory requirements of the two methods in the moderately high-dimensional case with $D = 20000$. For a modest 2-layers network and employing Float32 weights (each requiring 4 Bytes (B) for storage), the memory needed to store the parameters amounts to $D^2 \times 4B \times 2(\text{layers}) = 3.2GB$. Assuming a minibatch size of 100, data and activations require around 10-100 MB which is clearly negligible. The computed gradients will require the same space as the parameters, raising the memory footprint to over 6GB. For the gradient computations themselves, our method doesn't require additional memory (theoretically), while explicit automatic differentiation requires storing as many jacobian terms as the size of the minibatch, thus requiring over 300GB in our simple case. As this is clearly unfeasible on common hardware accelerators, we can drop the parallelization of the jacobian terms computation to considerably reduce memory consumption (bringing it down to over 9GB in our case), but this comes at the cost of further slowing down an already inefficient procedure.

While the simple analysis above shows a clear advantage for our proposed method, from the practical point of view many additional technical details might play a role in incrementing the memory requirements of both methods (e.g. loading of libraries and computing environment, just-in-time compilation steps, intermediate computations that can't be fused together...). In figure 4 we report a simple profiling of the memory consumption of the two methods, which shows how the difference is relevant in practice.

## H.2 Relative gradient optimization behaviour with different optimizers

In this section we report some additional observations analyizing the relative gradient optimization behaviour with different optimizers.

Figure 5: 2D toy examples trained with SGD. True distribution on the left, predicted densities on the right.

Figure 6: Log-likelihood evolution on MNIST validation set.

In figures 5 and 6 we compare the optimization behaviour using vanilla Stochastic Gradient Descent (SGD) and Adam. Results on toy datasets like those in figure 2 in the main paper are shown in figure 5. It can be seen that the data densities are modeled convincingly. We also report (figure 6) the evolution of the loss with SGD and Adam on density estimation on MNIST. The two methods seem to reach convergence at comparable speed: SGD is faster initially, but in the longer run Adam appears to achieve a better performance faster. Ultimately, both methods achieve a comparably good result.

### H.3 Density estimation

**Architecture** Although mentioned all throughout the paper, let us recall the neural network used for these experiments. We here rely on the usual feedforward architecture, that is, a neural network for which the input is sequentially passed through an interleaving series of matrix multiplications and non-linear activation functions, being the last operation a matrix multiplication.

**Nonlinearities** Note that, since we make use of square weight matrices, the only two hyperparameters left in our architecture are the number of layers in the network, $L$, and the non-linearity used. We consider two types of non-linearities. First, a smooth version of the leaky-ReLU activation function with a hyperparameter $\alpha$,

$$s_L(x) = \alpha x + (1 - \alpha) \log(1 + e^x). \tag{34}$$

Second, a weighted sum of the identity and hyperbolic tangent functions with two hyperparameters, $\alpha$ and $\beta$, controlling the steepness and "level of linearity" of the activation function,

$$s_T(x) = \tanh(\alpha x) + \beta x. \tag{35}$$

However, in our experiments, these two hyperparameters for the $s_T$ nonlinearity are fixed to $\alpha = 1$ and $\beta = 0.1$ always. Both of these nonlinearities are relatively smooth, and while no closed form solution for their inverse is available they can be inverted easily with a Newton method; in practice, for our parameter choice, we use a fixed number of 100 iterations which seems to be (way) more than sufficient.

**Toy examples** For all the experiments shown in figure 2 of the main paper, we always use Adam as optimizer, fix the batch size and number of layers $L$ to 100, use biases, and fix the activation function to $s_L$ with $\alpha = 0.3$. We chose as base distribution (that is, the distribution of the latent variables) the standard normal distribution. We plot, as in the quantitative experiments, the best model found

during the training. Regarding the data, we sampled five-thousand samples for the training set and five-hundred points for the test set. The only changing hyperparameters across the figures is the learning rate and the number of epochs, which are summarised in table 2.

Table 2: Hyperparameters used for figure 2 of the main paper.

|  | MoG | half moons | sine |
|---|---|---|---|
| learning rate | 0.001 | 0.001 | 0.005 |
| no. of epochs | 2000 | 1300 | 4000 |

**Quantitative results on MNIST** To obtain the density results on the MNIST dataset, the same preprocessing as in [43] has been applied. Note that we do not include the contribution due to this preprocessing in the reported log-likelihood values. [8] For the model architecture, we fixed the number of layers to 2. Note that competing models reported in table 4 of the main paper are taken from [43] and employ a higher number of parameters. We used the smooth Leaky-ReLU (34) with $\alpha = 0.01$ and a standard normal distribution as a distribution for the latent variables. The optimization has been performed with Adam with default parameters. The hyperparameters search has been performed over learning rate values of $0.001, 0.0005, 0.0001$ and batch sizes of $10, 100$. For each run, we selected the model whose performance did not improve in the successive 30 epochs of training (i.e. we chose the model at epoch 10 if all the values of the loss for epochs 11 to 40 were higher then the value after 10 epochs). The best hyperparameters selection is shown in table 4.

**Convergence time on MNIST** To get an idea of the running time of our method in a real-world scenario, one epoch on MNIST ($D = 784$, 50k training samples) on a modern laptop CPU takes an order of tens of seconds, a $\sim 4.5\times$ speedup compared to "standard" optimization (which is roughly consistent with figure 3, which was obtained with a slightly different experimental setup) and $\sim 50\times$ speedup with respect to "autodiff". Our convergence time is $\sim 15$ min. While the speed-up is already visible at this data dimensionality, the difference is expected to be larger at higher dimensionality.

**Quantitative results** First, we want to remark that the data used for the experiments shown in table 1 was pre-processed in the exact same way as described in [43].

For the results shown in such table (MNIST excluded) a more exhaustive hyperparameter search has been performed. Particularly, for each dataset a grid-search was run with the options shown in table 3, taking for each experiment the model with best validation log-likelihood obtained during training and, across experiments, getting the one with best test log-likelihood. Experiments were again trained using Adam and, instead of fixing the number of epochs, training was finished with an early-stopping criteria that evaluates the validation set every 25 epochs and has a patience of 5 trials. The best hyperparameters selection is shown in table 4.

Table 3: Hyperparameters considered for the grid search.

|  | Option #1 | Option #2 | Option #3 |
|---|---|---|---|
| activation | $s_L, \alpha = 0.3$ | $s_L, \alpha = 0.01$ | $s_T$ |
| no. layers | 25 | 50 | 100 |
| learning rate | 0.001 | 0.0005 | 0.0001 |
| batch size | 10 | 50 | 100 |
| base distribution | standard normal | hyperbolic secant |  |
| bias | Yes | No |  |

Regarding the rest of the models shown in that table, we reproduce the exact same experiments as those described in [43]. Therefore, the considered models have the same architecture and stopping criteria as the ones shown in table 1 of the aforementioned paper. The only difference with respect to the results shown in table 1 of [43] and table 1 in our paper is the number of trainable parameters. As mentioned in section 5, in order to perform a fair comparison, we tweaked the hyperparameters of each architecture so they have approximately the same number of parameters.

Table 4: Hyperparameters for the results in table 1 in the main paper.

| | POWER | GAS | HEPMASS | MINIBOONE | BSDS300 | MNIST |
|---|---|---|---|---|---|---|
| activation | $s_L, \alpha = 0.3$ | $s_L, \alpha = 0.3$ | $s_L, \alpha = 0.3$ | $s_T$ | $s_T$ | $s_L, \alpha = 0.01$ |
| no. layers | 50 | 100 | 50 | 25 | 25 | 2 |
| learning rate | 0.001 | 0.001 | 0.001 | 0.0001 | 0.0001 | 0.0001 |
| batch size | 100 | 100 | 50 | 100 | 100 | 10 |
| base dist. | std normal | std normal | hyper. secant | std normal | hyper. secant | std normal |
| bias | Yes | Yes | No | Yes | No | Yes |

Specifically, we first trained our model as described above and, once we knew the number of parameters of the best-performing model (which is approximately $LD^2$) we used the formulae shown in table 3 of [43] to find to which values we should fix the hyperparameters $L$ and $H$ of their models so that they have the same number of parameters.

As a final remark, note that there is one degree-of-freedom in those equations (for every $L$ there is a value of $H$ solving the given equation). Therefore, for each of the considered models and datasets, we run two different experiments, one with $L = 1$ and another with $L = 2$ (as similarly done in [43]), finding afterwards the proper value of $H$ to match the number of trainable parameters of our best model for that same dataset.

## Footnotes

[8]We thank T. Anderson Keller and Emiel Hoogeboom for pointing this out.