[Reviews · NeurIPS 2020]

Review 1

Summary and Contributions: The paper proposes an improvement in the time complexity of computing maximum likelihood estimates for an invertible parametrized function mapping a latent variable to an observable variable. In the main setting, the inverse function is assumed to be computable by a complete deep neural net, and the distribution on the latent variable is "nice" (a coordinate-wise product distribution). The computation naturally requires computing the gradient of the loss function using back propagation. This naturally decomposes into a layer-by-layer computation. The bottleneck is the Jackobian term (the partial derivatives of the mapping with respect to the observables), which a straightforward analysis shows requires inverting the matrix of edge weights. For a D X D matrix, this is trivially D^3, and even using non-trivial methods much worse than D^2. The main contribution of the paper is to propose a back propagation update using relative gradients based on multiplicative perturbation of the matrix of weights. This leads to D^2 performance per layer, and the improvement is demonstrated by experiments.

Strengths: The paper clearly touches on very important questions: learning latent variable models, using deep networks to model anticipated data, and solving questions in unsupervised learning. The contribution is simple and elegant, and hence potentially applicable in practice. The improvement is mathematically provable.

Weaknesses: It would have been nice to see examples of concrete and rigorous high dimensional latent variable models that can be solved in this manner.

Correctness: The paper seems to be correct.

Clarity: The paper is clearly written.

Relation to Prior Work: As far as I can tell, they relate correctly to past work.

Reproducibility: Yes

Additional Feedback:


Review 2

Summary and Contributions: The paper proposes to use relative gradient, which simplifies the gradient of the Jacobian term, in optimization for fully connected layers in unsupervised learning. It claims the approach is less restricted compared to normalizing flows while still has similar performance improvement over naive gradient approaches.

Strengths: The paper has pointed to an interesting perspective of simplifying the gradient computation of the Jacobian term by using alternative optimization schemes. The proposed approach is simple and seems effective on the small set of evaluations. If the approach is really effective, it will have significant impact.

Weaknesses: The main limitations of the paper are two folds: 1) The formulation of relative gradient is somewhat hand-wavy, and lacks theoretical justification of its convergence properties. Even analysis on a simple problem setting is good enough to demonstrate at least that a) it converges, b) converges to the correct solution, c) some characterization of its convergence rate. 2) Evaluation is weak. In order to back up the claim, the approach should be evaluated on more challenging tasks with comparison to more alternatives, such as normalizing flow variants. In addition, one major advantage claimed by the paper is that relative gradient is less restrictive than normalizing flow approaches. To further back up the claim, the paper should identify a few example problems where normalizing flow is suffering practically and compare it with the performance of relative gradients.

Correctness: The derivation and empirical methodology look correct.

Clarity: The paper is clear. However, I think the paper spends too much efforts in introducing existing concepts and too little is on the proposed approach, for example, convergent guarantees/properties, more thorough empirical evaluations.

Relation to Prior Work: The discussion on prior work seems sufficient.

Reproducibility: Yes

Additional Feedback: In Table 1, what is the number reported? Please specify it explicitly in the table. Also what are the other models being compared to? Either explain them in the caption or in the main text, and if applicable, add citation near the model names. ========================================== After reading other reviews and author's rebuttal, I maintain the same rating.


Review 3

Summary and Contributions: Quite a bit of recent research on deep density estimation under the normalizing flows umbrella has focused on efficiently computing (a restricted form of) the Jacobian term that appears in the objective. Such models operate with a set of transformations where the computation of this term is easy. While arbitrary distributions can be learned by such methods, the features that are learned are quite skewered which can prevent learning a proper disentangled representation. This paper presents a conceptually simple method to optimize for exact maximum likelihood in such models. In particular, the authors consider a transform from the observed to the latent space which is parameterized by fully connected networks with the only constraint that the weight matrices are invertible. Since the parameters of the transformation are matrices, the authors use properties of Riemannian geometry of matrix spaces to derive updates in terms of the relative gradient. This method gives an efficient way to update the Jacobian term, not just in the context of normalizing flows, but in more general contexts where they might appear. The main model considered is given by equation (2), its factorized variant is given by equation (3), and a conditional variant by (4). Evaluation of the gradient of the first term in this is easy as long as the model can be considered to have a simple form for the latent density. The second term, which is the Jacobian term, represents the main bottleneck underlined above. It is known that forward-backward mode auto diff scales as O(LD^3), where L is the number of layers and D is the data dimension. This is because computing the gradient of the log-determinant involves a matrix inversion. The contribution of this paper is to use the relative gradient such that the computation of the matrix inversion is not necessary. Moreover, by applying the relative gradient carefully (equation 15), we also avoid an expensive matrix matrix multiplication that is needed to compute the relative gradient from the ordinary gradient. Thus, we get a cheap method to compute the gradient of the log-determinant term.

Strengths: The paper gives a very simple method to train linear flows using the relative gradient instead of the normal gradient.

Weaknesses: Since the main selling point of the method is computational speedup, it would be useful to report numbers that indicate so in the experiments.

Correctness: Yes.

Clarity: The paper is well written. There are occasional typos and incorrect capitalizations (like M for maximum likelihood), which the authors should remove.

Relation to Prior Work: Discussion of prior work is thorough.

Reproducibility: Yes

Additional Feedback:


Review 4

Summary and Contributions: The main contribution is to propose to perform gradient descent on a change of variable that simplifies computations, avoiding an expensive (cubic cost) matrix inverse, in a deep latent variable model. They derive the corresponding expressions for a fully connected network, and empirically compare their technique to gradient descent (using Adam, see comments below) achieving a substantial speedup. -- after author's rebuttal: I find the new experiment with SGD interesting, that seem to show that the proposed technique works as described in sec 4 (and not only in conjunction with adam) I however don't understand the author's statement that SGD is slower than Adam., where there plot shows the oppose during the first 50 iterations. Regarding autodiff vs standard I still don't understand the difference, since autodiff is supposed to implement the same formulas as in sec 3 (but automatically). This should be clarified in the main body. I still agree (like other reviewers) that the empirical part is weak. For these reasons I will keep my score unchanged

Strengths: The technical part is sound, and introduce a technique that, to the best of my knowledge, has not been used in previous work. Since it drastically reduce the computation cost of training a fully connected type deep latent variable model, it is certainly a significant contribution. In order to fully appreciate the significance of the technique, I wonder whether similar change of variable could be applied to other deep learning setups.

Weaknesses: On the empirical part, I find it striking that no experiment use regular SGD, without Adam. A confounding effect of Adam could be to somehow compensate for the missing matrix inverse in the update step. Did you try it without success? This is an important drawback since the theory is really elegant and sound, but it is not clear that the experiments support the claim. In Appendix F about including the bias term in the relative gradient expression, you mention that you use a projected gradient algorithm. Can you elaborate a bit about that ? Does this simply correspond to setting the elements of the augmented matrix to zero?

Correctness: While the technical part is correct as far as I can tell, I find the empirical evaluation rather weak since, as already mentioned, it misses a simpler experiment without Adam, that would empirically evaluate the proposed theory.

Clarity: The paper is well written, with everything clearly explained. Since a large amount of material is in the appendix, I would have appreciated direct references to the corresponding sections in the main body in order to improve readability. Some things that should be clarified: - in the main body what is the difference between standard and autodiff in fig 1? - in table 1, I assume that you report log likelihoods, but this is missing in the caption.

Relation to Prior Work: This is a rather uninformed opinion since I do not know the recent literature in the field, but is seems like the literature review is complete. NB Part of the related work section is in the appendix.

Reproducibility: Yes

Additional Feedback: typos: l114-115: with

[Author Response · NeurIPS 2020]

We thank the reviewers for their comments and the largely positive feedback. Reviewers agree that "*the paper clearly touches on very important questions*" (**R5**), since " *efficiently computing (a restricted form of) the Jacobian term*" has been the focus of "*quite a bit of recent research on deep density estimation*" (**R7**). Reviewers also praised the novelty and correctness of the contribution: it is a shared opinion that "*the paper has pointed to an interesting perspective of simplifying the gradient computation*" (**R6**). The improvement our approach provides "*is demonstrated by experiments*" and "*mathematically provable*" (**R5**); "*since it drastically reduces the computation cost of training a fully connected type deep latent variable model, it is certainly a significant contribution*" (**R8**). The contribution was praised as "*elegant*", "*conceptually simple*" (**R7**) and "*potentially applicable in practice*" (**R5**).

A comment which was made (**R5**, **R6**, **R8**) is that, given its wide applicability, it would be interesting to try the method on additional applications. While we agree, we focused on density estimation as it is one of the most challenging problems in probabilistic modeling, and on the theoretical and empirical characterization of the computational improvement, believing (and reviewers seem to agree with us, s.a.) that our work constitutes a significant contribution in these regards.

**R6**: *Rigorous formulation and convergence properties of relative gradient:* We will add more details on this. Rigorous theory of the relative gradient is well known [Absil et al. , *Optimization Algorithms on Matrix Manifolds*, 2009]; in our paper we propose a simple and accessible derivation. Our proposed method is a stochastic first order optimization algorithm on the manifold of invertible $D \times D$ matrices: almost sure convergence of the parameters to a critical point of the gradient of the cost function can be derived for such SGD with decreasing step size under suitable assumptions (e.g. [Bonnabel, *SGD on Riemannian manifolds*, 2013]). We will include these references in the paper.

**R6**: *Example problems where normalizing flows suffer compared to relative gradients:* We mention some in the paper, and will further elaborate on this. Most state-of-the-art flow models employ autoregressive transformations and/or coupling layers permuting the inputs between successive layers. These architectures have several limitations, e.g. they can not learn a properly disentangled feature representation. *Linear flows* provide a strict generalization thereof, and our approach can be used for their computationally efficient training. Alternative methods decompose the weight matrix $W$ into easier-to-optimize transformations. One alternative is to compute the $PLU$ decomposition of $W$ and optimize the $L$ and $U$ transformations. The drawback in this approach is that the permutation matrix $P$ cannot be learned. A more flexible alternative is to consider the $QR$ decomposition of $W$, however computing $Q$ in full generality requires $\mathcal{O}(D^3)$ operations, matching the complexity of the naive optimization of linear flows. An experimental comparison of the performance of the $PLU$ and $QR$ decompositions against the direct optimization of $W$ is in [Hoogeboom et al., *Emerging convolutions for generative normalizing flows*, 2019], describing numerical and stability issues when using $PLU$. We will include this discussion and reference in the paper.

**R6**: *Too much emphasis on existing concepts, too little on the proposed approach:* We will try to balance this.

**R7**: *Computation time in the experiments:* We will add more details to the paper. One epoch on MNIST ($D = 784$, 50k training samples) on a modern laptop CPU takes an order of tens of seconds, a $\sim 4.5\times$ speedup compared to "standard" optimization and $\sim 50\times$ speedup w.r.t. "autodiff" (see below). Our convergence time is $\sim 15$ min.

*Broader impact:* We will add a more thorough discussion of this point.

**R8**: *Experiments use regular SGD, without Adam:* Thanks for pointing this out. We will add experiments with standard SGD. In figure 1 below, we show results on toy datasets like those in figure 2 in the main paper. It can be seen that the data densities are modeled convincingly. We also report (figure 2 below) the evolution of the loss with SGD and Adam on density estimation on MNIST. Optimization with SGD appears to converge slower (confirming the notion that Adam is a more effective optimizer) but ultimately leads to a comparably good result. Similar considerations hold for all datasets in Table 1 in the main paper, with SGD performance being slightly worse but comparable.

**R8**: *More comments on the projected gradient algorithm:* The augmented matrix formalism allows the computation of the relative gradient for the biases. The projection step corresponds to zeroing out gradients on the last row of the augmented matrix, as remarked by the reviewer. We will report explicit formulas in the appendix.

**R6**, **R8**: *In Table 1, what is the number reported? What are the models being compared to?:* The reported numbers are log-likelihoods. The competing models are the same as in reference [34]. We will clarify this in the caption.

**R7**, **R8**: *In the main body what is the difference between standard and autodiff in fig. 1?:* "Standard" refers to computing gradients of the Jacobian term as explained in section 3 in the main paper; "autodiff" refers to computing the Jacobian term and its gradients via automatic differentiation with the Jax package. We will clarify this in the main text.

Figure 1: 2D toy examples trained with SGD. True distribution on the left, predicted densities on the right.

Figure 2: Log-likelihood evolution on MNIST validation set.

[Meta-Review · NeurIPS 2020]

The focus of the work is deep density estimation (also called normalizing flows). Particularly, the authors focus on the generative model x=f(s) as defined in (1) where the observation (x) is described as the invertible non-linear function (f) of a latent variable (s). They take a maximum-likelihood perspective (2) where g_{\theta}, the approximation of the inverse of f, is the composition of g_1=\sigma_1(W_1 \cdot), ..., g_L=\sigma_L(W_L \cdot) invertible and differentiable component functions. They propose to use the relative gradient method to optimize \theta to speed up computations. Deep density estimation is an important problem in machine learning. While the proposed relative gradient approach is widely-applied for instance in the independent component analysis literature, the reviewers agreed that its adaptation to the deep density estimation task is interesting and can be of practical interest.